# Transcriptomics Dissection of Calorie Restriction and Exercise Training in Brown Adipose Tissue and Skeletal Muscle

**DOI:** 10.3390/nu15041047

**Published:** 2023-02-20

**Authors:** Yonghao Feng, Zhicheng Cui, Xiaodan Lu, Hongyu Gong, Xiaoyu Liu, Hui Wang, Haoyu Cheng, Huanqing Gao, Xiaohong Shi, Yiming Li, Hongying Ye, Qiongyue Zhang, Xingxing Kong

**Affiliations:** 1Department of Endocrinology, Jinshan Hospital, Fudan University, Shanghai 201508, China; 2State Key Laboratory of Genetic Engineering, School of Life Sciences, Fudan University, Shanghai 200438, China; 3Precision Medicine Center, Jilin Province General Hospital, Changchun 130021, China; 4School of Life Sciences, Inner Mongolia University, Hohhot 010000, China; 5Institute of Metabolism and Integrative Biology, Fudan University, Shanghai 200438, China; 6Department of Endocrinology and Metabolism, Huashan Hospital, Fudan University, Shanghai 200040, China; 7State Key Laboratory of Genetic Engineering, Department of Endocrinology and Metabolism, Huashan Hospital, School of Life Sciences, Fudan University, Shanghai 200438, China

**Keywords:** calorie restriction, exercise training, brown adipose tissue, skeletal muscle, extracellular matrix

## Abstract

Calorie restriction (CR) and exercise training (EX) are two critical lifestyle interventions for the prevention and treatment of metabolic diseases, such as obesity and diabetes. Brown adipose tissue (BAT) and skeletal muscle are two important organs for the generation of heat. Here, we undertook detailed transcriptional profiling of these two thermogenic tissues from mice treated subjected to CR and/or EX. We found transcriptional reprogramming of BAT and skeletal muscle as a result of CR but little from EX. Consistent with this, CR induced alterations in the expression of genes encoding adipokines and myokines in BAT and skeletal muscle, respectively. Deconvolution analysis showed differences in the subpopulations of myogenic cells, mesothelial cells and endogenic cells in BAT and in the subpopulations of satellite cells, immune cells and endothelial cells in skeletal muscle as a result of CR or EX. NicheNet analysis, exploring potential inter-organ communication, indicated that BAT and skeletal muscle could mutually regulate their fatty acid metabolism and thermogenesis through ligands and receptors. These data comprise an extensive resource for the study of thermogenic tissue molecular responses to CR and/or EX in a healthy state.

## 1. Introduction

Skeletal muscle and brown adipocytes have been found to share a common Pax7^+^/Myf5^+^ lineage, and transcriptional profiles of their precursor cells showed similarities [1,2]. Both skeletal muscle and brown adipose tissue (BAT) use energy substrates, such as glucose and fatty acids, to generate heat and to maintain core body temperature [3]. Consistent with this, skeletal muscle and BAT have been shown to have synergistic effects in response to cold challenge [3]. Skeletal muscle has multiple mechanisms of heat production involving shivering and non-shivering thermogenesis [4,5]. BAT encodes uncoupling proteins to dissipate the mitochondrial proton gradient and produce heat [6,7]. Nevertheless, the transcriptomic adaptations of these two tissues to various physiological treatments (e.g., calorie restriction (CR) and exercise) are not fully understood. 

CR, a classical example of negative energy balance, extends healthy lifespan from yeast to mammals [8,9,10]. Though the precise mechanisms of reaction to CR are still not fully defined, the metabolic benefits have been studied in different tissues. For example, short-term CR feeding enhances skeletal muscle stem cell function, mitochondrial function and muscle repair and, eventually, improves muscle insulin sensitivity [11]. It is also evident that CR can improve liver lipid metabolism and reduce systemic inflammation [12,13]. Given these findings, it is clear that adaptation to CR leads to progressive recruitment of metabolic tissues, but effects on BAT are still poorly explored. While a previous study demonstrated that CR led to browning of white adipose tissue [14], another study indicated that CR during pregnancy diminished thermogenic capacity in the offspring, including impaired BAT sympathetic innervation and thyroid hormone signaling [15]. It would thus be interesting to investigate the molecular changes in BAT in response to CR.

Exercise training (EX) causes many adaptations in the body that contribute to beneficial effects on health, including enhancing insulin sensitivity and reducing circulating lipid concentrations, primarily via adaptations to skeletal muscle [16,17,18]. Recently, studies have begun to address exercise-induced adaptations in adipose tissue [19]. While exercise is reported to induce white adipose tissue browning [20,21,22], the effects of EX on BAT are controversial. Different results have been obtained in various studies investigating BAT mitochondrial activity and gene expression, glucose uptake, the lipidome of BAT and the thermogenesis of BAT after acute and chronic exercise. Several rodent studies determined that EX increases mitochondrial activity and thermogenesis in rodent BAT [18,23,24,25], while others showed decreased mitochondrial gene expression and thermogenesis [26,27]. In humans, studies have shown that exercise reduces glucose uptake in BAT [27,28]. Given that the biological analysis is debated, the molecular pathway analysis still needs to be clarified. Collectively, CR, EX, or a combination of both are accepted as effective strategies in obesity prevention and treatment.

Enhanced skeletal muscle or BAT activities play essential roles in the treatment of obesity. Skeletal muscle can produce cytokines, including proteins, peptides, enzymes and metabolites, and these cytokines can contribute to the weight loss induced by EX [29,30]. In addition, evidence has also shown that skeletal muscle-induced cytokines can induce browning of WAT in response to both CR and EX [14,31]. Therefore, we hypothesized that CR and EX may yield greater benefits for weight loss treatments. Although the effect of CR on changes in metabolites in BAT has been reported [32], the signaling pathway at the transcriptome level for skeletal muscle and BAT in response to CR and EX is unclear.

Thus, in this study, we carried out RNA-seq of BAT and skeletal muscle to investigate the molecular and pathway alterations in response to CR and/or exercise interventions. To explore whether CR and EX have synergistic benefits, we restricted the EX mice to 70% calorie intake.

## 2. Materials and Methods

### 2.1. Experiment Model and Subject Details

Four-week-old healthy male C57BL/6J mice were purchased from GemPharmatech Co., Ltd. All mice were housed under standard laboratory conditions (12 h on/off) and in a temperature-controlled environment (22–24 °C) in the SPF Animal Research Center of Shanghai University of Sports (SYXK 2014-0002). The mice were given ad libitum access to food and water or fed a CR diet beginning at 8 weeks of age. The glucose fed to mice was tested three days before they were sacrificed. Mice were sacrificed 24 h after the last bout of EX. Mice in all groups were fasted for 6 h prior to tissue harvesting. Tissues were then rapidly dissected and processed or stored for analysis. 

### 2.2. CR

Compared to the baseline food intake, the food intake of the mice was reduced by 10% per week to a final level of 70% of the ad libitum food intake at the 8th week. 

### 2.3. Chronic Treadmill Exercise Training

Eight-week-old male C57BL/6J wild-type mice with and without CRs were randomly divided into the sedentary and EX groups. The treadmill treatment included 2 days of adaptive training. For the adaptive EX, the treadmill was set at a 10° incline and began with a 5 min 0 m/min acclimation period, followed by 10 m/min for 10 min and 14 m/min for another 10 min. Mice started to run on the third day (5° incline, 12 m/min for 1 h, 5 days a week for 8 weeks). Mice were fasted for 3 h prior to EX. 

### 2.4. Body Composition Measurement

A MiniSpec MQ10 nuclear magnetic resonance analyzer (Bruker) was used to measure the body composition of the mice according to the manufacturer’s instructions. Briefly, mice were put in a nuclear magnetic resonance tube and loaded in the machine. The body composition was measured automatically by the machine.

### 2.5. H&E Staining

BAT tissues were fixed in a 4% paraformaldehyde solution for at least 24 h and embedded in paraffin. Hematoxylin and eosin (H&E) (hematoxylin—E607317-0500, eosin—E607321-0100; Sangon Biotech, Shanghai, China) staining was undertaken. The tissues were processed as per routine for paraffin embedding, and 5 μm thick sections were cut and placed on glass slides. The paraffin-embedded sections were dewaxed with xylene, washed with a gradient of ethanol to water and then incubated with hematoxylin and eosin (Servicebio, Wuhan, China) for 4 min and sealed after conventional ethanol dehydration. Finally, the sections were analyzed under a Nikon light microscope at the indicated magnification. ImageJ software was used to calculate the areas of droplets in H*&*E images.

### 2.6. Bulk mRNA Sequencing

A total of 15 BAT samples, including 3 from the sedentary group (control), 4 from the CR group, 4 from the EX group (EX) and 4 from the CR + EX group (CREX), and 14 gastrocnemius muscle tissue samples, including 4 from the control group, 3 from the CR group, 4 from the EX group and 3 from the CREX group, were used to perform transcriptome analysis. The Trizol method was used to extract total RNA according to the manufacturer’s instructions. RNA quality was measured using an Agilent 2100 Bioanalyzer (RNA 6000 Nano Kit; Agilent Technologies, Santa Clara, CA, USA). A previously reported method was utilized to construct cDNA libraries for each sample [33]. Libraries were sequenced on a BGIseq500 platform (BGI-Shenzhen, China) using 150 bp paired-end reads aimed at 30 million reads per sample.

### 2.7. Sequence Alignment and Gene Expression Analysis

The raw sequencing data were filtered with trim-galore (v0.6.7) by removing reads containing sequencing adapters and reads with low-quality bases. The clean reads were mapped to the reference genome (mm10) using STAR (v2.7.10a). Quantification of gene expression was calculated using rsem (v2.0.1) with the following command: (rsem-calculate-expression --paired-end -p 20 –alignments samples.bam mm10). We conducted a differently expressed (DE) genes analysis for the two groups. Subsequently, differential expression analysis for the groups was carried out using DESeq2 (v1.38.0) with an adjusted *p* value < 0.05 and |log2FC| > log2(1.5). Then, GO and KEGG pathway analyses of the DEGs were carried out using the enrichGO and enrichKEGG functions in the R package clusterProfiler (v4.6.0).

### 2.8. Adipokine and Myokine Analysis

Based on previous secretome profiling data from adipose tissue and skeletal muscle obtained with the LC-MS/MS method, we determined the DEGs encoding adipokines and myokines. Then, we performed GO and KEGG analyses to explore the roles of these DEGs using the enrichGO and enrichKEGG functions in the R package clusterProfiler (v4.6.0) 

### 2.9. Deconvolution

We input gene counts from the bulk RNA-seq data from BAT and GAS, as well as the multi-subject single-cell profiles from previously published single-cell RNA sequencing (scRNA-seq) or single-nuclei RNA sequencing (snRNA-seq) data (skeletal muscle: GSE183288; BAT: GSE207705), into the MuSiC2 algorithm in R4.2.2. Cell types from the scRNA-seq and snRNA-seq were based on prior categorizations. Genes in the bulk RNA-seq data and single-cell and single-nuclei profiles used gene symbols as their identifiers. MuSiC2 used cell-type-specific gene expression from scRNA-seq data to characterize cell type compositions in the bulk RNA-seq. The estimated proportions were normalized to sum to 1 across the included cell types. The Wilcoxon test was used to calculate comparisons between group levels. 

### 2.10. BAT and Skeletal Muscle Tissue Crosstalk

To analysis ligand activity in the two tissues, we first defined the set of potentially active ligands in the ‘‘sender’’ tissue (i.e., skeletal muscle or BAT). Ligand–receptor interactions and weighted-networks data were downloaded from https://zenodo.org/record/3260758 (accessed on 8 January 2023). Ligands regulated by CR with or without EX for which at least one specific receptor was expressed (average mean expression over all conditions > 1 tag/kilobase (kb)) in the receiver tissue (i.e., BAT or skeletal muscle) were used for the next analyses. Then, NicheNet (v1.1.1) was used to rank the ligands based on how well they predicted whether a gene was linked with a gene set of interest compared to the background gene set. The gene sets of interest for the BAT and skeletal muscle were defined as metabolic-related DEGs. All other genes expressed in the receiver tissue (average mean expression over all conditions > 1 tag/kb) were considered background. Ligand activity scores were calculated as the Pearson correlation coefficient for the ligand–target regulatory potential scores of each selected ligand and the target indicator vectors, which indicated whether a gene belonged to the gene set of interest or not. For the top five ligands with the highest ligand activity, the corresponding receptors were exhibited in a ligand–receptor heatmap. Furthermore, the most prominent target genes for the top five ligands were chosen according to the regulatory potential score, with the genes presented relating to the indicated gene set of interested and being among the 5000 most strongly predicted targets of at least one of the top five ligands. Ligand–target gene interactions were illustrated in a circle plot using the R-package circlize (v0.4.15). 

### 2.11. Quantification and Statistical Analysis

Unpaired one-tailed t tests or Wilcoxon rank sum and signed rank tests were performed to compare the two groups. One-way ANOVA and post hoc Tukey’s multiple comparison tests were performed for intergroup comparisons of more than two groups.

## 3. Results

### 3.1. Phenotypic Response to CR and/or EX and Profiling of Two Metabolic Tissues

We studied 8-week-old C57BL/6J male mice treated with CR and/or treadmill running exercise training (EX) interventions for 8 weeks (Figure 1A; n = 38 across four groups) and then collected BAT and gastrocnemius muscle for transcriptomic profiling. Phenotypically, CR decreased body weight (BW) gain (Figure 1B). The reduced weight was because of the lower fat mass in the CR group compared to the control group, with no differences in lean mass (Figure 1C,D). The BAT weight was also lower in CR feeding mice (Figure 1E). EX had no significant effects on BW gain or body mass (Figure 1B–D and Appendix A). The level of fed glucose was lower in the CR and/or EX groups compared to the control group (Figure 1F). Moreover, CR decreased the size of lipid droplets in BAT, whereas EX enlarged the size (Figure 1G,H). Nonetheless, both CR and EX increased browning, showing synergistic effects in the browning of inguinal white adipose tissue, for which adipocytes are usually multilocular, and decreasing lipid droplet size (Appendix A). 

### 3.2. BAT-Level Gene and Pathway Alterations upon CR with or without EX

To investigate the molecular alterations, we conducted transcriptomic analysis of BAT under the CR and/or EX conditions. The principal component analysis plots of the different interventions are shown in Figure 2A,B. A total of 1362 differentially expressed genes (DEGs) were identified in the CR group compared to the control group, including 499 upregulated and 863 downregulated genes (Figure 2C and Appendix A). Additionally, a total of 781 DEGs were identified in the CREX group compared to the EX group, including 310 upregulated and 471 downregulated genes (Figure 2D and Appendix A). To gain further insights into these DEGs, we conducted gene ontology (GO) term and Kyoto Encyclopedia of Genes and Genomes (KEGG) pathway analyses. Upregulated DEGs induced by CR with or without EX were mainly enriched in lipid and small molecule metabolism, whereas downregulated DEGs were engaged in general functions and pathways related to the extracellular matrix (Figure 2E–H). These results indicated that CR and CREX might have similar effects on BAT (Figure 2I).

Next, we evaluated the similarities and differences between CR and CREX with regard to transcriptomic alterations. There were 222 upregulated and 341 downregulated genes overlapping in the CR and CREX comparisons. While 522 downregulated and 277 upregulated genes were only found in the CR comparison, 130 downregulated and 88 upregulated genes were found in the CREX comparison (Figure 2J). Analysis of pathways revealed that co-upregulated DEGs were mainly enriched in functions associated with the lipid metabolic process, while co-downregulated DEGs not only participated in the metabolic process but were also involved in cell motility (Appendix A). In CR-specific upregulated DEGs, the metabolic process and biosynthetic process were significantly enriched, and downregulated DEGs were primarily involved in cytoskeleton and immune-related functions (Appendix A). Upregulated DEGs in the CREX group were related to biosynthesis of amino acids and carbon metabolism, and downregulated DEGs in the CREX group related to general functions concerning the extracellular matrix (ECM) (Appendix A).

### 3.3. BAT-Level Gene Alteration upon EX with or without CR

Transcriptome analysis demonstrated that EX caused a few genes to change, which was consistent with the lack of changes in BW. PCA plots of the EX vs. control groups and CREX vs. CR groups are displayed in Appendix A. In total, 13 upregulated and 24 downregulated genes were found in the EX group compared to control (Appendix A), and 25 upregulated and 5 downregulated genes were found in the CREX group vs. the CR group (Appendix A).

### 3.4. Analysis of DEGs Encoding Adipokines upon CR with or without EX

BAT can secrete various adipokines executing autocrine, paracrine and endocrine regulatory functions [34]. We explored the effects of CR and/or EX on genes encoding adipokines [35,36,37]. In total, 111 different adipokine genes were profiled in the CR group compared to the control group, including 67 upregulated and 44 downregulated genes (Figure 3A), and 76 different adipokine genes were identified in the CREX group compared to the EX group, including 40 upregulated and 36 downregulated genes (Figure 3B). There were 21 upregulated and 32 downregulated adipokines overlapping in the two comparisons (Figure 3C). Pathway analysis showed that co-upregulated adipokines were mainly enriched in functions associated with amino acids, fatty acids and small molecule metabolic processes, while co-downregulated adipokines primarily participated in the cell cytoskeleton, cell migration, ECM–receptor interactions and protein digestion and absorption (Figure 3D,E). Moreover, CR-specific upregulated adipokines were related to small molecular metabolic processes, while CR-specific downregulated adipokines were associated with general functions related to the cell cytoskeleton, migration and immune response (Figure 3F,G). As only a few CREX-specific adipokines were identified, we did not analyze the CREX-specific pathways.

### 3.5. Skeletal Muscle-Level Gene and Pathway Alterations upon CR with or without EX

To profile transcriptional alterations in skeletal muscle response to CR and/or EX, bulk RAN-seq of skeletal muscle was performed. The principal component analysis plot for the four-group intervention is shown in Figure 4A,B. A total of 1308 DEGs were identified in the CR group compared to the control group, including 593 upregulated and 715 downregulated genes (Figure 4C and Appendix A). Additionally, a total of 1454 DEGs were determined in the CREX group compared to the EX group, including 588 upregulated and 866 downregulated genes (Figure 4D and Appendix A). Pathway analysis demonstrated that all upregulated DEGs induced by CR were mainly enriched in the cholesterol metabolism function, whereas downregulated DEGs were engaged in general functions and pathways related to the ECM and ECM–receptor interactions (Figure 4E,F), similar to the findings for BAT. Upregulated DEGs induced by CREX were mainly enriched in the general function associated with the ribosome, whereas downregulated DEGs were engaged in general functions and pathways related to the ECM, ECM–receptor interaction and oxidative phosphorylation (Figure 4G,H).

Next, we assessed the similarities and differences in transcriptomic alterations caused by CR and CREX (Figure 4I). There were 290 commonly upregulated and 434 commonly downregulated genes in the CR and CREX comparisons (Figure 4J). Pathway analysis revealed that co-upregulated DEGs were enriched in functions relating to the AMPK signaling pathway, neuroactive ligand–receptor interactions and cellular senescence, while co-downregulated DEGs were involved in functions relating to the inner mitochondrial membrane protein complex, cell differentiation, cell adhesion and cell migration (Appendix A).

Then, we analyzed the role of CR-specific regulated DEGs. Upregulated DEGs participated in the lipid metabolic process and the biosynthesis of amino acids, and downregulated DEGs were primarily involved in ECM–receptor interactions, insulin resistance and lipid metabolism (Appendix A). CREX-specific upregulated DEGs were related to the ribosome, and CREX-specific downregulated DEGs were related to general functions concerning channel activity and lipid metabolism (Appendix A).

### 3.6. Skeletal Muscle-Level Gene and Pathway Alterations upon EX with or without CR

Similarly to the BAT transcriptome, a few DEGs were identified in skeletal muscle upon EX with or without CR. The principal component analysis plots of the EX versus control groups and CREX versus CR displayed no significant differences (Appendix A). DEG analysis distinguished 48 DEGs in the EX group versus the control group, encompassing 20 upregulated genes and 28 downregulated genes (Appendix A), and 15 DEGs in the CREX group versus the CR group, consisting of 6 upregulated genes and 9 downregulated genes (Appendix A). GO and pathway analyses verified that the upregulated DEGs induced by EX were related to circadian rhythm, whereas the downregulated DEGs were related to the PPAR signaling pathway (Appendix A).

### 3.7. Analysis of DEGs Encoding Myokines upon CR with or without EX

Skeletal muscle has been reported to secret myokines to communicate with other tissues [38,39,40,41]. Therefore, we investigated the effects of CR and/or EX on genes encoding myokines. In total, 100 different myokine genes were profiled in the CR compared to control groups, including 35 upregulated and 65 downregulated myokines (Figure 5A), and 112 different myokine genes were determined in the CREX group compared to the EX group, including 30 upregulated and 82 downregulated myokines (Figure 5B). There were 9 upregulated and 44 downregulated adipokines overlapping in the two comparisons (Figure 5C).

GO and pathway analyses showed that co-upregulated myokines were mainly enriched in functions associated with calcium ion transport and the ECM, while co-downregulated DE myokines were primarily related to cell growth, cell migration, cellular response to environmental stimuli and cancer-related pathways (Figure 5D,E). CR-specific upregulated DE myokines were related to vitamin binding, lipid location, carbon metabolism and the ECM, and downregulated DE myokines were linked to general functions concerning skeletal system development, cell motility, immune response and the MAPK signaling pathway (Figure 5F,G). CREX-specific upregulated DE myokines were associated with general functions concerning the ribosome, and downregulated DE myokines were related to various metabolic processes, immune response and the lipid and atherosclerosis signaling pathway (Figure 5H,I). As EX had little effect on transcriptome alterations, it only caused tiny changes in genes encoding myokines.

### 3.8. Cell Proportion Alterations resulting from CR and/or EX across the Two Tissues

BAT and skeletal muscle are heterogeneous tissues with different cell types. We analyzed the cell proportions in BAT and skeletal muscle. To recognize the cell population compositions of the two tissues, we performed deconvolution analysis using published single-cell or single-nuclei RNA sequencing data (BAT, GSE207706; skeletal muscle, GSE183288). Statistical comparisons were not performed for proportions of cell subgroups close to or equal to zero. For the BAT, five cell populations—adipose cells, myogenic cells, mesothelial cells, endothelial cells and smooth muscle cells—were analyzed (Figure 6A). Only mesothelial cell proportions were decreased in the CR group compared to control. The other four cell proportions did not show significantly different responses to CR without or with EX. In addition, EX decreased the adipose cell proportions and increased the myogenic and endogenic cell proportions. For skeletal muscle, deconvolution analysis identified ten cell populations: muscle fiber cells, satellite cells, fibro-adipogenic progenitor (FAP) cells, B cells, dendritic cells, monocyte cells, neutrophil cells, smooth muscle cells, endothelial cells and glial cells (Figure 6B). CR with or without EX had no significant effects on muscle fiber cell proportions. CR probably increased the satellite and endothelial cell proportions but decreased immune cell proportions, such as dendritic and neutrophil cell proportions (Figure 6B). While endothelial cell proportions were increased by EX, immune cell (neutrophil and dendritic cells) proportions were decreased. Both CR and EX decreased smooth muscle cell proportions. CR had no effects on the proportions of the other three cell populations with or without EX.

### 3.9. The Crosstalk between BAT and Muscle upon CR with or without EX

BAT and skeletal muscle originate from the same precursors in the somites and display multifaceted interactions. Our group previously reported that BAT could communicate with skeletal muscle via myostatin [1]. Others have shown that skeletal muscle secreted myokines to regulate the metabolism of adipose tissue [20,21,22]. We analyzed the networks between BAT and skeletal muscle. Gene networks linking selected DEGs from the two tissues encoding interacting proteins were constructed. Biologically meaningful modules of interacting proteins linked with metabolism were shown (Figure 7A). Considering that the main cell compositions of BAT and skeletal muscle were adipocyte and muscle fiber cells, respectively, we performed NicheNet analysis to explore potential inter-tissue communication between BAT and skeletal muscle. We found that CR-regulated, BAT-derived ligands had the potential to modulate skeletal muscle-selective gene programs related to fatty acid metabolism and thermogenesis. The five BAT ligands with the highest predicted activity in CR were Apoe, Bmp5, Nptn, Col5a3 and Inhbb. All of the five had corresponding receptors with high interaction potential in skeletal muscle (Figure 7B). These receptors could regulate several target genes in skeletal muscle correlated with fatty acid metabolism and thermogenesis (Figure 7C). Of the top five potential ligands in BAT, Nptn was upregulated, and Inhbb, Col5a3, Apoe and Bmp5 were downregulated (Figure 7D). A total of 15 target genes regulated by these receptors were identified, of which Ehhadh, Acsl1, Hadha and Hadhb were downregulated; Hsd17b7 was upregulated; and Acadvl, Hadh, Acaa2, Acadl, Ndufs1, Dhcr24, Cox5a, Ndufb2, Atp5h and Fdft1 were not significantly different in the CR group compared to the control group (Figure 7E).

Next, we investigated whether CR-regulated, skeletal muscle-derived ligands had the potential to regulate BAT-selective gene programs related to fatty acid metabolism and thermogenesis (Figure 7F). The top five skeletal muscle ligands with the highest predicted activity in the CR group were Apoe, Col5a3, Gpi1, Angpt1 and Col4a1, and all had corresponding receptors with high interaction potential in BAT (Figure 7F). These receptors could regulate multiple target genes in BAT correlated with fatty acid metabolism and thermogenesis (Figure 7G). Of the top five potential ligands in skeletal muscle, Apoe and Gpi1 were upregulated and Angpt1, Col4a1 and Col5a3 were downregulated (Figure 7H). A total of seven target genes regulated by these receptors were identified, of which Hsd17b7, Dhcr24 and Fdft1were upregulated and Ndufs1, Cox5a, Atp5h and Ndufs8 were not significantly different in the CR group compared to the control group (Figure 7I).

Furthermore, we performed NicheNet analyses of the CREX and EX conditions. The top five BAT-derived ligands with the highest predicted activity in the CR with EX condition were Apoe, Bmp5, Col5a3, Pdgfd and Sfrp2, and all had several receptors with high interaction potential in skeletal muscle (Appendix A). These receptors could regulate multiple target genes in skeletal muscle correlated with fatty acid metabolism and thermogenesis (Appendix A). The five skeletal muscle-derived ligands with the highest predicted activity in the CREX group were Gpi1, Col4a1, Icam2, Cx3cl1 and Nampt, and all had several receptors with high interaction potential in BAT (Appendix A). These receptors could regulate multiple target genes in BAT correlated with fatty acid metabolism and thermogenesis (Appendix A). Taken together, the findings suggested that Apoe and Col5a3 were mainly involved in interactions under the different conditions, indicating that Apoe and Col5a3 might play vital roles in thermogenesis and fatty acid metabolism.

## 4. Discussion

Skeletal muscle and BAT are two well-described thermogenic sites that utilize distinct mechanisms for heat production. CR and EX are two lifestyle interventions that aim to produce improvements in metabolic health. The comparative effects of EX vs. CR on transcriptomics in BAT vs. skeletal muscle have not previously been reported. In the present study, we compared the effects of 8 weeks of CR with/without EX on mRNA expression and signaling pathways relating to thermogenesis, metabolism and ECM; in addition, we examined the cell populations and potential ligand–receptor communications between BAT and muscle. 

Our data showed that CR and CREX can significantly reduce body weight and fat mass; however, we did not observe significant differences between the EX and control groups. CR, EX and CREX could decrease fed glucose; however, there was a greater decrease in the CREX group compared to the EX group. Our findings highlight that strategies targeting obesity should combine both CR and EX. White adipose tissue, accounting for the largest volume of adipose tissue in humans, is critical for energy storage, endocrine communication and glucose homeostasis [42]. Browning is a process that plays a critical role in white adipose tissue regulation and leads to increased thermogenesis [43]. CR has been reported to result in browning of white adipose tissue in lean male C57BL/6 and BALB/c mice after only 1 week of restriction [14], which is consistent with our results. In a previous study, CR enhanced the thermogenesis of BAT by affecting the tricarboxylic acid cycle and fatty acid degradation [32]. Inhibition of the ECM and fatty acid metabolism have been reported to be linked to thermogenesis of BAT [44,45]. In our study, CR showed a decline in the ECM but enhanced fatty acid biosynthesis in BAT, which also indicated that CR activates thermogenesis in BAT. It is possible that, during CR, BAT switches function to store lipids, as shown in 9 month old rats subjected to 6 months of CR and ECM remodeling [46,47]. Unlike BAT, CR induced cholesterol metabolism and decreased ECM in skeletal muscle. The gene expression level of the ECM, which includes many collagen genes, coincides with changes in muscle size, increasing during muscle hypertrophy [48] and decreasing during experimental muscle atrophy [49]. Alternatively, changes in ECM gene expression may simply reflect changes in protein turnover.

EX and CR can reduce weight loss by increasing skeletal muscle and BAT burning; however, evidence has shown that exercise alone without dietary restrictions cannot reduce body weight significantly, which indicates that exercise alone is not effective for weight loss, especially for those patients with cardiometabolic diseases [50]. Our study aimed to investigate the joint effects of CR and EX on RNA transcription in skeletal muscle and BAT. Our data showed that the targeted pathway of weight loss in skeletal muscle and BAT was the metabolism of fatty acid, which is consistent with the function of skeletal muscle and BAT. Our data highlighted that strategies for weight loss should consider combining CR and EX simultaneously. In line with a previous study [11], our data showed that satellite population was increased, especially in the CREX group, indicating that CREX-induced metabolic factors play an important role in regulating stem cell function. In addition, consistent with Cara’s study [32], our results showed that, compared with EX, the amino acid biosynthesis pathway was upregulated in CREX, which indicated that CR plays a critical role in amino acid metabolism. 

Unexpectedly, the present study did not detect significance in the ability of EX to alter mRNA levels of markers of lipid or carbohydrate metabolism in BAT or skeletal muscle. Several reasons may account for this. First, the time of tissue collection: mice were sacrificed 24 h after the last bout of running when the mRNA profiles were probably back to normal. Second, the intensity of exercise: we trained the mice with a low-intensity recipe, which might not have been sufficient to cause gene expression alterations. Third, the duration of running: the mice only ran 1 h per day. Longer duration and higher intensity for the exercise are required in further investigations.

Our results in Figure 2 and Appendix A (BAT) and Figure 4 and Appendix A (skeletal muscle) showed that the difference in genes alteration between the CR and control groups was substantially stronger than the difference between the EX and control groups, which suggested that CR and EX may exert distinct, nonoverlapping and frequently additive effects on BAT and skeletal muscle. Further work is needed to systematically isolate targetable pathways and address the resulting interactions to develop optimal strategies to promote healthy benefits. 

## Figures and Tables

**Figure 1 nutrients-15-01047-f001:**
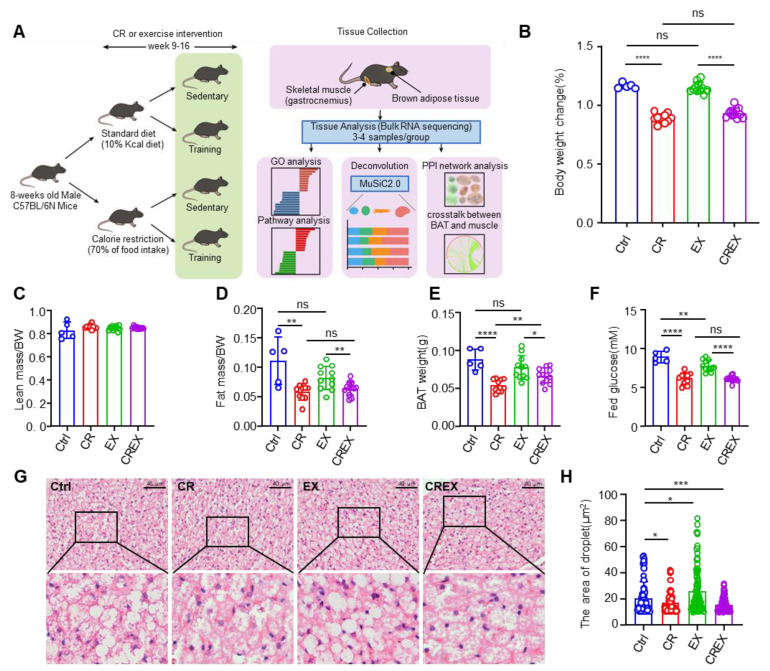
Study overview and phenotypic responses. (**A**) Overview of the mouse study and tissue profiling. (**B**–**F**) Body weight change (**B**), ratio of lean mass to body weight (**C**), ratio of fat mass to body weight (**D**), BAT weight (**E**) and fed glucose (**F**) in the four intervention groups. (**G**) Representative hematoxylin and eosin staining images of BAT (scale bars, 40 μm). (**H**) Areas of the droplets from H&E staining images of BAT. Statistical comparisons were carried out using unpaired one-tailed t tests or Wilcoxon rank sum and signed rank tests (**B**–**F**,**H**). Data are represented as means ± SD. * *p* < 0.05, ** *p* < 0.01, *** *p* < 0.001,**** *p* < 0.0001; ns, not significant; BW, body weight; CR, calorie restriction; EX, exercise training; CREX, calorie restriction combined with exercise training; BAT, brown adipose tissue.

**Figure 2 nutrients-15-01047-f002:**
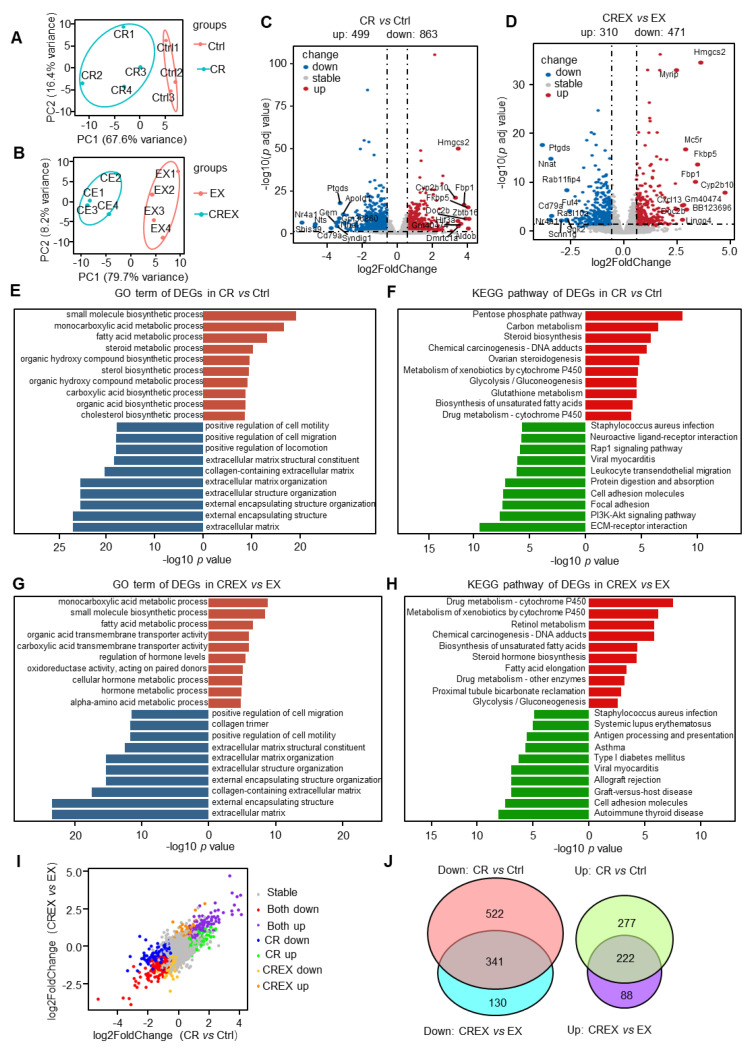
Transcriptomic changes in BAT upon CR with or without EX. (**A**,**B**) Principal component analysis plot of the BAT samples in the control and CR groups (**A**) and in the EX and CREX groups (**B**). (**C**,**D**) Volcano plots of DEGs in the CR vs. control groups (**C**) and in the CREX vs. EX groups (**D**). The top 20 genes with the highest absolute values for the log2FC were labeled (down: *p* < 0.05 and log2FC < −0.58; up: *p* < 0.05 and log2FC > 0.58). (**E**) Enriched GO terms for all DEGs in BAT upon CR. (**F**) Enriched KEGG pathways for all DEGs in BAT upon CR. (**G**) Enriched GO terms for all DEGs in BAT upon CREX. (**H**) Enriched KEGG pathways for all DEGs in BAT upon CREX. (**I**) Dot plot of genes’ log2FC in CR vs. control groups and CREX vs. EX groups. Stable, genes that were not differently expressed in the CR vs. control groups or CREX vs. EX groups; Both down, downregulated DEGS across the two comparisons; Both up, upregulated DEGS across the two comparisons; CR down, specific downregulated DEGS in the comparison between the CR and control groups; CR up, upregulated DEGS in the comparison between the CR and control groups; CREX down, specific downregulated DEGS in the comparison between the CREX and EX groups; CREX up, upregulated DEGS in the comparison between the CREX and EX groups. (**J**) Venn plot of common and distinct DEGs upon CR with or without EX. Brick red, enriched GO terms for upregulated DEGs; navy blue, enriched GO terms for downregulated DEGs; bright red, enriched KEGG pathways for upregulated DEGs; green, enriched KEGG pathways for downregulated DEGs.

**Figure 3 nutrients-15-01047-f003:**
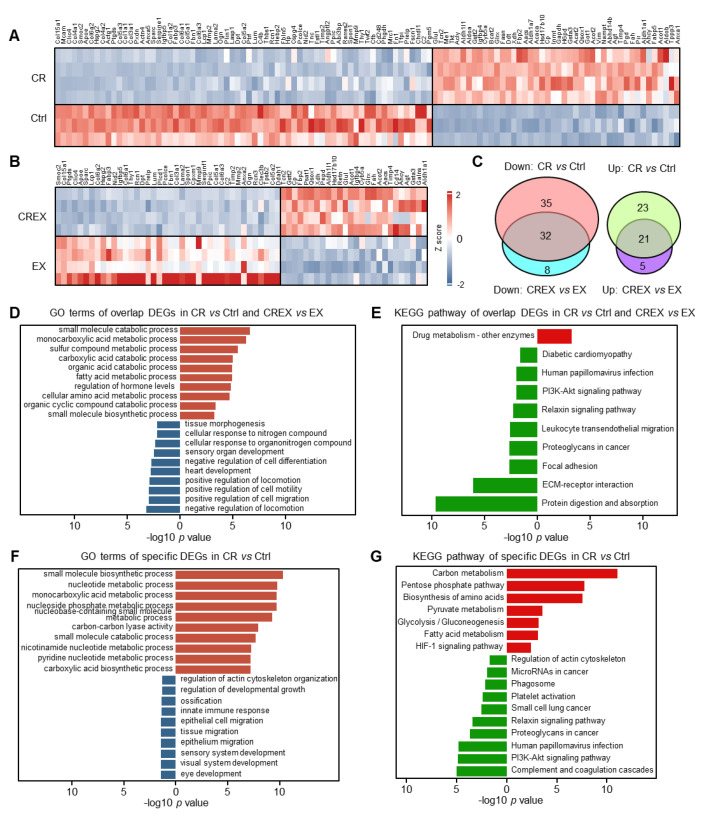
GO terms and KEGG pathways enriched by DEGs encoding adipokines upon CR with or without EX. (**A**,**B**) Heatmap plot of DEGs encoding adipokines in CR vs. control groups (**A**) and in CREX vs. EX groups (**B**). (**C**) Venn plot of common and distinct DEGs encoding adipokines upon CR with or without EX. (**D**) Enriched GO terms for common DEGs encoding adipokines in BAT upon CR with or without EX. (**E**) Enriched KEGG pathways for common DEGs encoding adipokines in BAT upon CR with or without EX. (**F**) Enriched GO terms for specific DEGs encoding adipokines in BAT upon CR. (**G**) Enriched KEGG pathways for specific DEGs encoding adipokines in BAT upon CR. Brick red, enriched GO terms for upregulated DEGs encoding adipokines; navy blue, enriched GO terms for downregulated DEGs encoding adipokines; bright red, enriched KEGG pathways for upregulated DEGs encoding adipokines; green, enriched KEGG pathways for downregulated DEGs encoding adipokines.

**Figure 4 nutrients-15-01047-f004:**
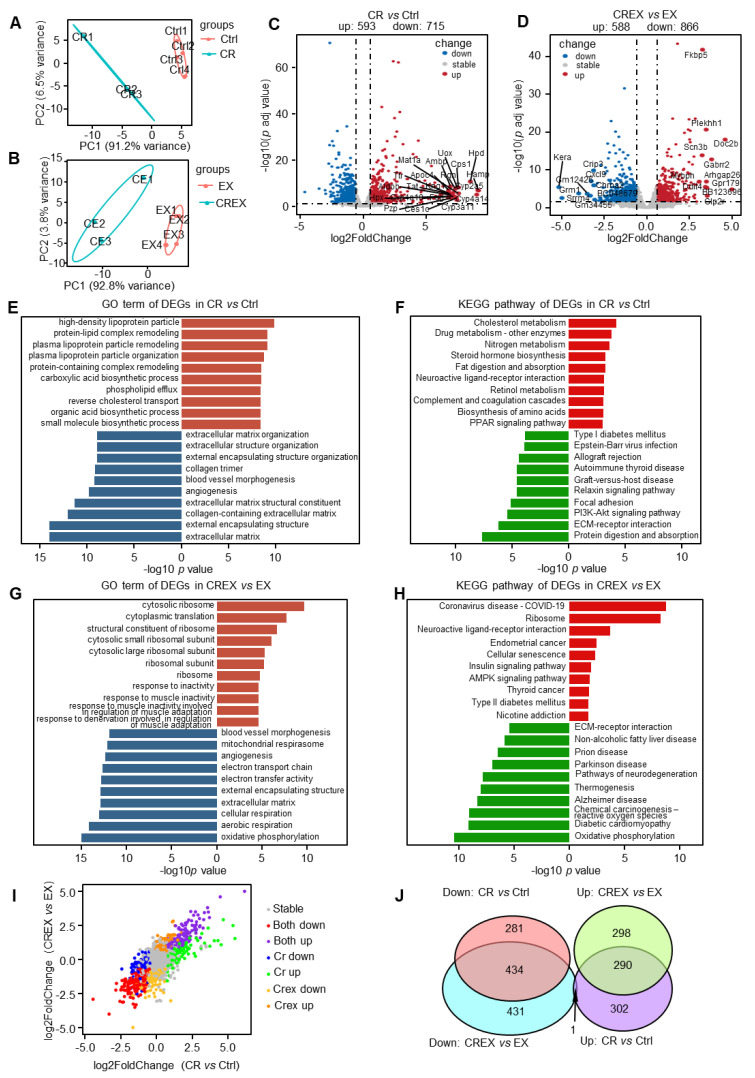
Transcriptomic changes in skeletal muscle upon CR with or without EX. (**A**,**B**) principal component analysis plot of the skeletal muscle samples in the control and CR groups (**A**) and in the EX and CREX groups (**B**). (**C**,**D**) Volcano plots of DEGs in the CR vs. control groups (**C**) and in the CREX vs. EX groups (**D**). The top 20 genes with the largest absolute values for log2FC were labeled (down: *p* < 0.05 and log2FC < −0.58; up: *p* < 0.05 and log2FC > 0.58). (**E**) Enriched GO terms for all DEGs in the skeletal muscle upon CR. (**F**) Enriched KEGG pathways for all DEGs in the skeletal muscle upon CR. (**G**) Enriched GO terms for all DEGs in the skeletal muscle upon CREX. (**H**) Enriched KEGG pathways for all DEGs in the skeletal muscle upon CREX. (**I**) Dot plot of genes’ log2FC in CR vs. control groups and CREX vs. EX groups. Dots with different colors represent genes in different patterns. Stable, genes that were not differently expressed in the CR vs. control groups or CREX vs. EX groups; Both down, downregulated DEGS across the two comparisons; Both up, upregulated DEGS across the two comparisons; CR down, specific downregulated DEGS in the comparison between the CR and control groups; CR up, upregulated DEGS in the comparison between the CR and control groups; CREX down, specific downregulated DEGS in the comparison between the CREX and EX groups; CREX up, upregulated DEGS in the comparison between the CREX and EX groups. (**J**) Venn plot of common and distinct DEGs upon CR with or without EX. Brick red, enriched GO terms for upregulated DEGs; navy blue, enriched GO terms for downregulated DEGs; bright red, enriched KEGG pathways for upregulated DEGs; green, enriched KEGG pathways for downregulated DEGs.

**Figure 5 nutrients-15-01047-f005:**
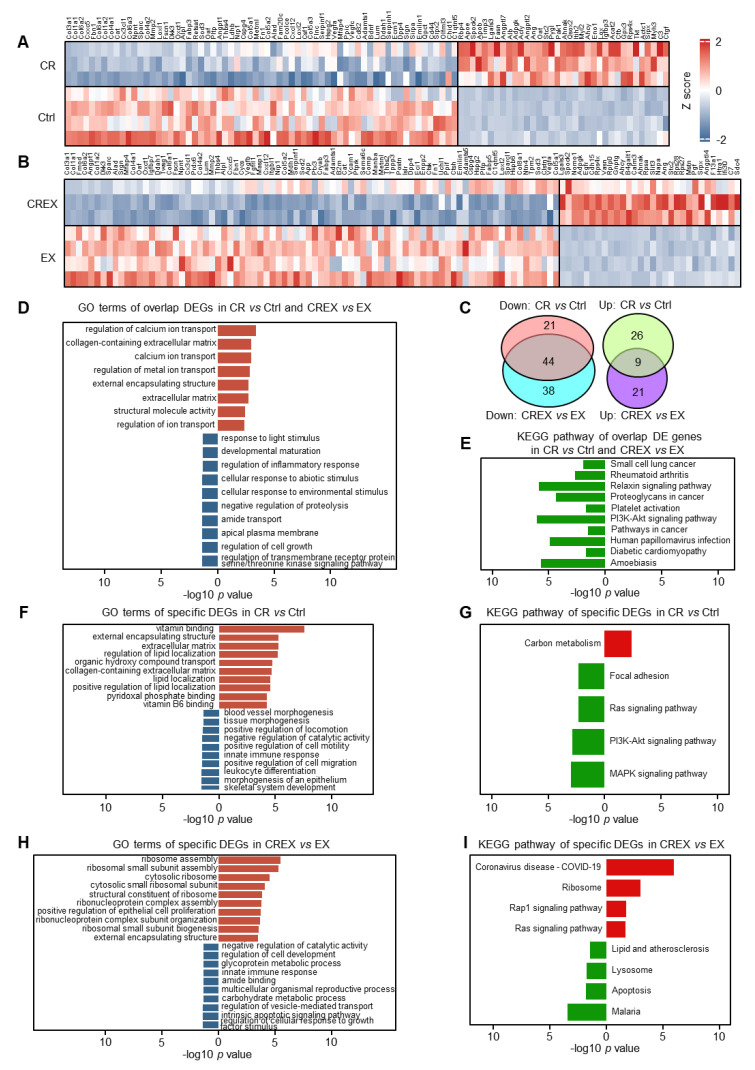
GO terms and KEGG pathways enriched for DEGs encoding myokines upon CR with or without EX. (**A**,**B**) Heatmap plots of DEGs encoding myokines in the CR vs. control groups (**A**) and in the CREX vs. EX groups (**B**). (**C**) Venn plot of common and distinct DEGs encoding myokines upon CR with or without EX. (**D**) Enriched GO terms for common DEGs encoding myokines in the skeletal muscle upon CR with or without EX. (**E**) Enriched KEGG pathways for common DEGs encoding myokines in the skeletal muscle upon CR with or without EX. (**F**) Enriched GO terms for specific DEGs encoding myokines in the skeletal muscle upon CR without EX. (**G**) Enriched KEGG pathways for specific DEGs encoding myokines in the skeletal muscle upon CR. (**H**) Enriched GO terms for specific DEGs encoding myokines in the skeletal muscle upon CR with EX. (**I**) Enriched KEGG pathways for specific DEGs encoding myokines in the skeletal muscle upon CR with EX. Brick red, enriched GO terms for upregulated DEGs encoding myokines; navy blue, enriched GO terms for downregulated DEGs encoding myokines; bright red, enriched KEGG pathways for upregulated DEGs encoding myokines; green, enriched KEGG pathways for downregulated DEGs encoding myokines.

**Figure 6 nutrients-15-01047-f006:**
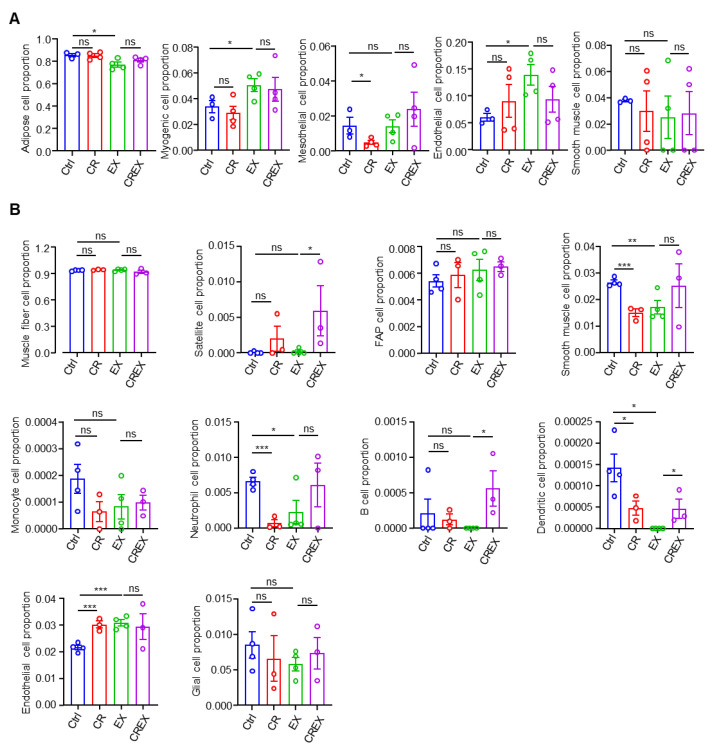
Cell proportion alterations as a result of CR with or without EX across the two tissues. (**A**) Boxplots of estimated cell type proportions in BAT bulk RNA sequencing from four intervention groups. Cell populations with numbers equal to or close to zero are not shown. (**B**) Boxplots of estimated cell type proportions in skeletal muscle bulk RNA sequencing from four intervention groups. Cell populations with numbers equal to or close to zero are not shown. Statistical significance was measured using unpaired one-tailed t tests or Wilcoxon rank sum and signed rank tests. Data are represented as means ± SEM. * *p* < 0.05, ** *p* < 0.01, *** *p* < 0.001; ns, not significant. FAP, fibro-adipogenic progenitor.

**Figure 7 nutrients-15-01047-f007:**
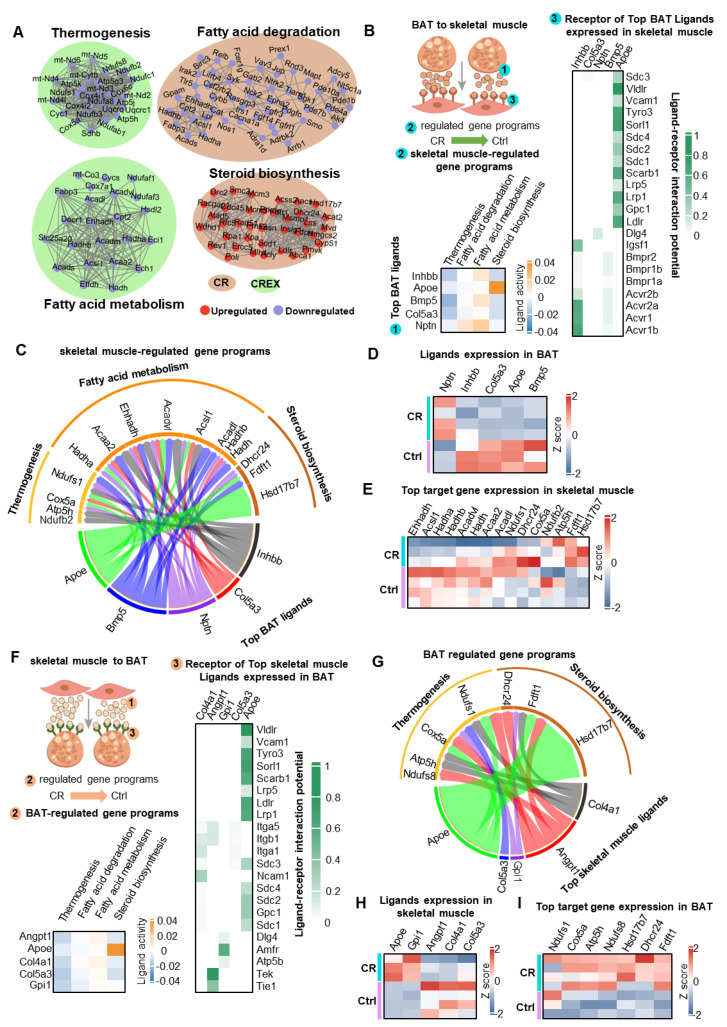
The crosstalk between BAT and muscle. (**A**) Gene networks for selected DEGs from the two tissues that encoded interacting proteins clustered by MCODE, with each cluster named according to the most significantly enriched pathway. The clusters are colored according to the DEG direction and tissue category. (**B**) NicheNet workflow (left top), the activity of the top five putative BAT ligands (left bottom) and the interaction potential for the putative receptors of the top five BAT ligands in skeletal muscle (right). (**C**) Predicted interactions between BAT ligands and their predicted skeletal muscle target genes associated with the indicated KEGG pathways. (**D**,**E**) Expressions of the genes encoding the top five predicted upstream ligands in BAT (**D**) and of their top target genes in skeletal muscle related to the indicated KEGG pathways (**E**). (**F**) NicheNet workflow (bottom), the activity of the top five putative skeletal muscle ligands (bottom) and the interaction potential for the putative BAT receptors (right). (**G**) Predicted interactions between skeletal muscle-derived ligands and their predicted BAT target genes belonging to the indicated KEGG pathways. (**H**,**I**) Expressions of the genes encoding the top five predicted ligands in skeletal muscle (**H**) and of their top target genes in BAT associated with the indicated KEGG pathways (**I**).

## Data Availability

The accession no. for the RNA-Seq data reported in this article is GEO: GSE222163. Other data generated or analyzed during this study are included in the published article and the Appendix A. All bioinformatics software used in the study is publicly available. Further information and requests for resources and reagents should be directed to, and will be fulfilled by, Xingxing Kong (kongxingxing@fudan.edu.cn).

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
