# Peer review of "Transcriptomics Dissection of Calorie Restriction and Exercise Training in Brown Adipose Tissue and Skeletal Muscle"

_nutrients, 2023, doi:10.3390/nu15041047_

Round 1
Reviewer 1 Report
In this paper, Feng Y et al. have described transcriptional profiling of BAT and SkM from mice treated with CR and/or EX. Moreover, the crosstalk between BAT and SkM in the same conditions was explored. Though the authors have reported interesting findings that need to be furtherly deepened but that however represent a staring points to future research, they should make a minor revision, taking into account the following issues.
-line 180. The meaning of iWAT should be explained and a brief description of WAT, BAT and browning should be added to Introduction or Results paragraph.
-Figure S1C. The result on browning shown in the this image should be quantified to be considered.
-3.2, 3.5 and 3.9 paragraphs. Figure S2A-S2B, S4A-S4B and S6A should be moved from supplemental Figures to Figure 2, 4 and 7 respectively. Indeed, starting a result paragraph with supplemental results is not confortable to read.
-line 212. Could Figure 2J refer instead to Figure S2J?
-line 284. Could Figure S4D refer instead to Figure S4F?
-line 288. Could Figure S4F, 4G refer instead to Figure S4H, S4G?
-line 290. Could Figure S4H, 4I refer instead to Figure S4I, S4J?
-line 358. What is endogenic cell?
-Discussion paragraph could be improved both explaining the main results obtained and stressing their relevance.
Reviewer 2 Report
The paper presents research on transcriptomics dissection of the calorie restriction and exercise training in skeletal muscle.
The manuscript is well written and the results of interest.
Important strengths include the detailed investigations.
The paper could nicely fit in the Special Issue “Adipose Tissue Metabolism and Exercise in Health and Disease”.
Title: omit the /or
Title: Avoid abbreviations in the title. Instead, the abbreviated terms need to be explained.
In general, the manuscript uses a lot of abbreviations that confuse reading.
The physiological mechanisms at work could be elaborated in more detail.
Are the hypothesized mechanisms identical in humans? What can we learn from mice research for human physiology?
In humans likely more factors can confound the results, e.g. motivation, stress, etc. This needs to be considered in mice research as well to create a comparable setting.
Reviewer 3 Report
This work is devoted to study the transcriptional response from muscles and brown adipose tissue to moderate calorie restriction and moderate exercise. This is a modern research that sheds light on the molecular mechanisms that underlie improved physical condition and counteract the development of metabolic diseases. The authors tried to evaluate not only individual responses from these organs, but also the molecular mechanisms of their possible interaction. The authors chose an adequate model of the experiment, used modern methods of bioinformatic analysis, and presented extensive well-illustrated results. However, in my opinion, a major revision of the manuscript is required for publication.
Major remarks.
Introduction.
In my opinion, the introduction is not sufficiently clear to understand scientific background on the basis of which these studies were conducted. I would like to understand:
1. Why were these 2 tissues chosen to evaluate the transcriptional response to CR and Ex? Only on the basis that they have common cellular precursors? It is clear that these 2 tissues produce heat, but what are the similarities or functional interactions in response to CR and Ex between them? What is known in the literature about this?
2. What assumption is tested during the execution of this study.
3. What aim did the authors set for themselves.
Discussion
In my opinion, the discussion is written too concisely. The conclusions presented in the last paragraph (L 456-459) and the conclusions presented in the Summary (L 26-28) in no way follow from this discussion.
The authors write:
"In the present study, we compared the effects of 8 weeks of CR with/ without EX on mRNA expression and signaling path ways in thermogenesis, metabolism, and ECM; in addition, we examined the cell populations and potential ligands-receptors communications between BAT and muscle". L435-438
However, they do not further interpret the results obtained and do not discuss, for example, the question of how the ligands-receptor communications discovered by them can be associated with the transcriptional response they have identified and which cell populations can be involved in these responses. Since the authors did not present any hypothesis and did not formulate the purpose of the study, it remains unclear why these studies were carried out, what the study provided for understanding the interaction between muscles and BAT in response to CR and EX, and what contribution these molecular responses can make to maintaining health and counteracting the development of metabolic diseases (L433-434).
Minor remarks
How many animals were in the groups? According to fig. 1 there were more than 4. How mice were selected for sampling for RNA analysis in the BAT and muscles. Were they the same animals?
BAT of what localization was taken for analysis?
How the size of fat droplets was calculated (L178)
How browning (L179-180) in iWAT was estimated?
What does "fad glucose" (L 177) mean if control animals were not fed within 6 hours before to be sacrificed?
Why did only the control animals not receive food within 6h, and what about the others?
L 173 figure 1B shows not BW gain, but BW
L 46 “precise mechanisms of CR are still not fully defined”
My be: precise mechanisms of reaction to CR are still not fully defined (?)
L 23 “CR induces genes encoding adipokines or myokines alteration in BAT and SkM, respectively”
My be: CR induces alterations in expression of genes encoding adipokines or myokines in BAT and SkM, respectively (?)
Round 2
Reviewer 3 Report
This work is devoted to study the transcriptional response from muscles and brown adipose tissue to moderate calorie restriction and moderate exercise. This is state-of-the-art research that sheds light on the molecular mechanisms that underlie improvement of health under negative energy balance. The authors tried to evaluate not only individual responses from these organs, but also the molecular mechanisms of their possible interaction. The authors chose an adequate model of the experiment, and presented a large, well-illustrated work in the Results section. They assessed the effects of calorie restriction, exercise, and their combined effects on weight of body, white and brown fat, blood glucose levels, transcriptional response and composition of cell population in BAT and muscles, and gene expression of ligands and their receptors in these tissues. The authors used modern methods and gave a brief interpretation of the results obtained in the Results section. This work is of undoubted interest and contributes to the understanding of the interaction between the two tissues involved in the energy expenditure in the form of heat in response to a negative energy balance: a decrease in energy intake (calorie restriction) and an increase in energy expenditure during exercise. Undouble, this work should be published, but, in my opinion, it is not ready for publication in its present form.
The Introduction and Discussion sections should be improved as they do not follow the rules for journal authors. The rules for authors state
Introduction: The introduction should briefly place the study in a broad context and highlight why it is important. It should define the purpose of the work and its significance, including specific hypotheses being tested. The current state of the research field should be reviewed carefully and key publications cited. Please highlight controversial and diverging hypotheses when necessary. Finally, briefly mention the main aim of the work and highlight the main conclusions. Keep the introduction comprehensible to scientists working outside the topic of the paper.
However, in my opinion, the introduction is not sufficiently clear to understand scientific background on the basis of which these studies were conducted. I could not find the formulation of the hypothesis and the purpose of the work in the revised manuscript. It is not clear, why BAT and muscles were chosen for the study, what is known about the interaction between these tissues during calorie restriction and exercise, and why authors evaluated not only separate, but also the combined effect of CD and Ex on the studied parameters. In addition, it was mistake in citation in the introduction (L57-58, [15]). The authors answered my questions, but did not include those answers in the introduction.
Response 1: Calorie restriction or regular exercise or a combination of the two is accepted as an effective strategy in preventing or treating obesity [1]. skeletal muscle can produce cytokines (In response to CR and Ex?), e.g., proteins, peptides, enzymes, and metabolites, which contributes to the CR and Ex induced weight loss (Is this your assumption, or you can confirm this with references?). Further, the cytokines in response to CR and Ex can induce browning of BAT from WAT (can you confirm this with references?), which plays a critical role in weight loss[1]( This paper [1] says nothing about WAT browning and cytokines and their role in weight loss). Hence, we have chosen these 2 tissues in our present study.
As I understand, authors are suggesting that both CR and EX activate the production and secretion of cytokines by muscle tissue, which leads to browning of white adipose tissue. But authors are studying the interaction between muscles and BAT not WAT. What is known about the effect of calorie restriction on brown adipose tissue function? The interpretation of this article (L57 Pico’s group reported that CR diminished the thermogenic capacity, including impaired BAT sympathetic innervation and thyroid hormone signaling [15]) is wrong. In this paper, it was shown that calorie restriction during pregnancy causes these effects in the offspring. The offspring themselves were not subjected to calorie restriction.
However, there is evidence in the literature regarding the effect of calorie restriction on biochemical changes in BAT (10.1093/gerona/glz023). How do these data correlate with your results? (This may be related to discussion)
Reference:
1. Magkos, F.; Hjorth, M.F.; Astrup, A. Diet and exercise in the prevention and treatment of type 2 diabetes mellitus. Nat Rev Endocrinol 2020, 16, 545-555, doi:10.1038/s41574-020-0381-5.
Point 2: What assumption is tested during the execution of this study.
Response 2: We assumed that CR+EX could exert synergic effect in brown adipose tissue and skeletal muscle.
Point 3: What aim did the authors set for themselves.
Response 3: We aimed to investigate whether CR+EX could have synergic effects on brown adipose tissue and skeletal muscle.
Despite the remarks made, the authors did not make any changes in the introduction in the revised manuscript.
Discussion Instruction for authors:
Discussion: Authors should discuss the results and how they can be interpreted in perspective of previous studies and of the working hypotheses. The findings and their implications should be discussed in the broadest context possible and limitations of the work highlighted. Future research directions may also be mentioned. This section may be combined with Results
Discussion was improved, but this improvement is not sufficient. The authors did not discuss results related to the effect of CR and Ex on body weight, white and brown fat, blood glucose levels, cell populations in BAT and muscle, and how these results might be related to transcriptional response and ligand-receptor interactions between brown fat and muscles. Since the hypothesis was not stated, it was not discussed whether the results obtained confirm or refute the hypothesis.
L472-476 You have shown that CR induced browning of WAT, and it is in line with results of other authors. The next sentence: “However, CR showed a decline in the ECM but enhanced fatty acid biosynthesis” It is not clear what tissue is being referred to in this context. If it was BAT, perhaps it should be discussed that WAT and BAT may differently response to CR.
L 500 Together, our results conclusively demonstrate that CR and EX exert distinct, nonoverlapping and frequently additive effects in BAT and skeletal muscle. Please, indicate, what results support this conclusion.
